# Polyphenols Content, Antioxidant Properties and Allergenic Potency of Organic and Conventional Blue Honeysuckle Berries

**DOI:** 10.3390/molecules27186083

**Published:** 2022-09-18

**Authors:** Alicja Ponder, Katarzyna Najman, Mateusz Aninowski, Joanna Leszczyńska, Agnieszka Głowacka, Agnieszka Monika Bielarska, Marius Lasinskas, Ewelina Hallmann

**Affiliations:** 1Department of Functional and Organic Food, Institute of Human Nutrition Sciences, Warsaw University of Life Sciences, Nowoursynowska 159c, 02-776 Warsaw, Poland; 2Institute of Natural Products and Cosmetics, Lodz University of Technology, Stefanowskiego 2/22, 90-924 Łódź, Poland; 3Cultivar Testing, Nursery and Gene Bank Resources Department, The National Institute of Horticulture Research, Konstytucji 3, 96-100 Skierniewice, Poland; 4Warsaw Department of Burns, Plastic and Reconstructive Surgery, Military Institute of Medicine, Szaserów 128, 04-141 Warsaw, Poland; 5Agriculture Academy, Department of Agrobiology and Food Sciences, Vytautas Magnus University, Donelaicio St. 58, 44248 Kaunas, Lithuania; 6Agriculture Academy, Bioeconomy Research Institute, Vytautas Magnus University, K. Donelaičio Str. 58, 44248 Kanuas, Lithuania

**Keywords:** organic, conventional, blue honeysuckle, phenolic compounds, flavonoids, anthocyanins, antioxidant activity, allergenic potency

## Abstract

Blue honeysuckle berries are a rich source of polyphenols with strong antioxidant properties. The purpose of this research was to determine the effect of organic and conventional cultivation on the polyphenols, antioxidant and allergenic potency of blue honeysuckle berry cultivars: ‘No 30’, ‘Jolanta’ and ‘Indygo’ in two growing seasons. Identification of individual polyphenols was performed using the HPLC method; the total polyphenols content and antioxidant activity were determined by spectrophotometric methods. The determination of allergic potency was tested by ELISA. In the second year of the study the total polyphenols were significantly higher in organic blue honeysuckle than in the conventional blue honeysuckle. In both growing seasons, the ‘Indygo’ cv. was characterized by the highest concentration of all bioactive compounds 3241.9 mg and 3787.2 mg per 100 g^−1^ D.W. A strong correlation was found between the polyphenol content and the antioxidant activity for organic fruit in both years, as well as for allergenic potency. Contrary to the best bioactive properties was ‘Indigo’ cv., with the highest allergenic potency (108.9 and 139.2 ng g^−1^ D.W.). The lowest content of specific allergens was found in the ‘No 30’ cv. Since honeysuckle is still a new cultivated plant, information about its allergenic potency is insufficient.

## 1. Introduction

Blue honeysuckle berry (*Lonicera caerulea* L. var. *kamtschatica* Sevast.) is a species belonging to the *Caprifoliaceae* family, including approx. 200 species, widespread mainly in the subtropical and temperate regions of the Northern Hemisphere. Currently, these fruits are widely cultivated in China, Japan, Canada and Russia [1]. They are relatively new to the Polish market, but the interest in these fruits is growing among consumers [2].

In botanical terms, blue honeysuckle berries are fleshy, elongated, bottle-shaped, multi-seed pseudo-berries, characterized by their purple–navy colour with a dark blue waxy coating, about 1 cm in diameter, 5 cm long and weighing about 3 g. They have a juicy, aromatic, sour–sweet flesh reminiscent of the black forest berry flavor [3]. In the literature, blue honeysuckle berries are known as ‘haskap’, ‘blue honeysuckle’, ‘kamchatka honeysuckle’, ‘edible honeysuckle’, ‘honeyberry’, ‘haska’, ‘zhimolost’, in terms of region, origin, variety, or cultivar [4,5].

Blue honeysuckle berries are a rich source of polyphenols and flavonoids and have antioxidant properties [6], containing valuable bioactive compounds that have shown pro-health, bacteriostatic and anti-inflammatory effects. Anthocyanins, flavonols and phenolic acids were also found in these berries [7]. Their quantity is largely dependent on the genotypes of individual cultivars, flowering time, climatic conditions, geographical location of cultivation, applied agrotechnical treatments, growing, degree of fruit ripeness, harvest period and storage conditions [5,8,9].

Blue honeysuckle berries can be a valuable component of the human diet due to the increase in dietary intake of their beneficial, biologically active compounds. Due to their rich chemical composition, high antioxidant value and health-promoting effects, these fruits have extremely high potential for use as ingredients in functional food products. Blue honeysuckle berries can be an important factor in preventing many chronic non-infectious diseases, such as cancer, obesity, diabetes and diseases of the cardiovascular system. Additionally, it is advisable to increase the use of the studied fruit species on a wider scale by the pharmaceutical and food industries. It is important to continue research to confirm the high health and utility value of blue honeysuckle berries [4,10,11,12]. Due to their easy cultivation, blue honeysuckle is usually grown under organic agriculture methods. There are a lot of data in the literature on the beneficial effects of organic cultiva-tion on the quality of berry fruits [13,14,15], but not on blue honeysuckle.

One of the main problems with food in the 21st century is allergenicity and food intolerance among consumers. In this situation, not all foods can be consumed by everyone [16]. Food allergy is a set of repeated symptoms of the immune system occurring in some sick people after ingestion of a substance (sometimes in trace amounts)—food allergenic proteins in sensitized individuals—which does not cause any symptoms in healthy people [17,18]. The occurrence of allergic reactions, both those of a mild nature (e.g., rash, urticaria) and those of a severe course (e.g., angioedema, respiratory failure, or anaphylaxis), is caused by IgE directed against specific food components, referred to as allergenic proteins [19], and the detection of these proteins is crucial for determining the potential level of food allergenicity [20]. On the other hand, food intolerance, also defined as non-allergic food hypersensitivity, is caused by non-allergic and non-immune mechanisms, disturbances in the metabolism of a specific food component, usually the lack of an enzyme, i.e., a substance causing its further transformation in the body (e.g., lactose intolerance—caused by a deficiency of the decomposing enzyme lactase) [21].

To diminish allergy symptoms, new cultivation technologies as well as fruit and vegetable cultivars are being promoted. According to current databases, the problem of food allergies is constantly growing and affects approx. 3–4% of adults and approx. 6% of children. The dominant allergy in Europe is Bet v1 (the main allergen of birch pollen). The homologues of Bet v1 (PR-10 family and profiling, Bet v2), also found in pollen from other trees and in many products of plant origin (e.g., apples, cherries, apricots, pears, strawberries), are similar in structure to Bet v1 up to 80% [22]. Although only 10% to 20% of pollen allergy patients are allergic to profilin, they react to a wide range of pollen from other plants and to foods of plant origin [23]. The multitude and prevalence of Bet v1-like allergens means that the number of potential cross-allergic reactions is enormous [22].

Many studies have shown that consumers with an allergy to strawberries could suffer allergies to other berry fruits. In contrast, some allergic reactions to strawberries are not typical allergies but represent a type of food intolerance [24]. Problems with allergic reactions to berry fruits are complicated and very difficult to understand. Some experiments have shown that organic fruits are safer than conventionally produced fruit for allergic individuals, but some have not [25,26]. Because blue honeysuckle is still a newly cultivated plant, especially in Poland, there is a lack of information about its allergenicity and food safety for consumers.

The aim of this study was to determine the effect of organic and conventional practices and cultivars on the polyphenol content, antioxidant status and allergy potency in three blue honeysuckle berry cultivars: ‘No 30’, ‘Jolanta’ and ‘Indygo’, in two growing seasons, i.e., 2018 and 2019.

## 2. Materials and Methods

### 2.1. Chemical Reagents

ABTS^+•^ (2,2′-azino-bis (3-ethylbenzothiazoline-6-sulfonic acid) (Sigma-Aldrich, Warsaw, Poland) Trolox (Sigma-Aldrich, Warsaw, Poland); acetic acid (99% pure for analysis, ChemPur, Warsaw, Poland); acetonitrile (HPLC pure, Sigma-Aldrich, Warsaw, Poland); alkaline phosphatase (Sigma-Aldrich, Warsaw, Poland); hydrochloric acid (35% pure for analysis, ChemPur, Piekary Śląskie, Poland); methanol (HPLC pure, Sigma-Aldrich, Warsaw, Poland); ethyl acetate (Sigma-Aldrich, Warsaw, Poland); aluminum chloride AlCl_3_ (Sigma-Aldrich, Warsaw, Poland); ortho-phosphoric acid (85%, HPLC, ChemPur, Piekary Śląskie, Poland); para-nitrophenyl phosphate (Sigma-Aldrich, Warsaw, Poland); phosphate-buffered saline (Sigma-Aldrich, Warsaw, Poland); potassium persulfate K_2_S_2_O_8_ (Sigma-Aldrich, Warsaw, Poland); sodium carbonate Na_2_CO_3_ (Sigma-Aldrich, Warsaw, Poland); Folin–Ciocalteu reagent (Sigma Aldrich, Warsaw, Poland); urotropine (Warchem, Poland); standards of phenolic acids (HPLC pure, Sigma-Aldrich, Warsaw, Poland): gallic, chlorogenic, caffeic, and *p*-coumaric acids; flavonoids (HPLC pure, Sigma-Aldrich, Warsaw, Poland): quercetin-3-*O*-rutinoside, quercetin-3-*O*-glucoside, kaempferol-3-*O*-glucoside, myricetin, luteolin, quercetin, kaempferol, cyanidin-3-*O*-glucoside, peonidin-3-*O*-glucoside, cyanidin-3-*O*-rutinoside; sodium hydroxyl peroxide (pure for analysis, ChemPur, Piekary Śląskie, Poland); and ultra-pure (deionized) water.

### 2.2. Origin of Fruits

The experiment was conducted in 2018 and 2019. Three blue honeysuckle berry cultivars, ‘No 30’, ‘Jolanta’ and ‘Indygo’, were chosen for analysis. The blue honeysuckle berries were cultivated in experimental orchards belonging to the Institute of Horticulture in Skierniewice (Poland). The organic orchard was located in Nowy Dwór-Parcela (51°51′55″ N 20°15′44″ E), and the conventional orchard in Dąbrowice (51°56′00″ N 20°06′10″ E) (separated by 16 km). Detailed information about the climate conditions is presented in Appendix A. Climatic data (min. and max. temperature and rainfall values) were obtained from a professional weather forecast measurement station located in Skierniewice (city of orchards), where the experiment was carried out. Sun hours per day information was obtained from the Institute of Meteorology and Water Management, National Research Institute. Each cultivar was represented by six bushes. One hundred grams of fruits were picked from each bush and mixed into one representative sample.

### 2.3. Preparation of Plant Material

The blue honeysuckle berries were harvested in the early morning and immediately transported to the laboratory. For each sample, 500 g of fruit were used, divided into two parts: one part was used for dry matter measurements, and the other was freeze-dried (Labconco freeze dryer (2.5), Warsaw, Poland, −40 °C, pressure 0.100 mBa). The freeze-dried samples were ground to fine powder (0.063 mesh) in a laboratory grinder (IKA^®^ A-11 basic), sealed and stored at −80 °C for further testing.

### 2.4. Dry Matter Content

Before freeze-drying, the content of dry matter (D.M.) using the weight method (Polish Norm PN-R-04013:1988) in the blue honeysuckle berries was determined. After weighing, empty glass-weighing vessels were filled with fresh honeysuckle berries, reweighed and then dried in a laboratory drier (FP-25 W, Farma Play, Poland) at 105 °C for 72 h. The dried samples were cooled to 21 °C and reweighed. The dry matter content was calculated for the blue honeysuckle berry samples based on their mass differences and given in units of 100 g^−1^ F.W. (fresh weight).

### 2.5. Total Polyphenols Measurement

Total polyphenols were measured by the modified colorimetric method; to measure the total polyphenols content of the tested samples, the colorimetric spectrophotometric method using Folin–Ciocalteu reagent was used.

To prepare the extracts, in plastic, sterile 50 mL tubes, 0.5 g (with an accuracy of 0.0001 g) of the freeze-dried plant material samples were weighed on an analytical balance (AS 220/X, Radwag, Radom, Poland), 40 mL of deionized water was added, vortexed for 60 s (2000 rpm) (Wizard Advanced IR Vortex Mixer, VELP Scientifica Srl, Usmate, Italy) for thorough mixing and then incubated in a shaking incubator (IKA KS 4000 Control, IKA^®^ Poland Ltd., Warsaw, Poland) for 2 h (50 °C, 250 rpm). After incubation, the samples were centrifuged in a refrigerated centrifuge (MPW-380 R, MPWMed. Instruments, Warszawa, Poland) for 15 min. (4 °C, 10.000 rpm), and the resulting clear supernatants were used to determine the total polyphenol content and the antioxidant activity (point 2.11).

To determine the total polyphenol content, 1.0 mL (respectively diluted aqueous sample extracts) was dosed into 50 mL volumetric flasks, and 2.5 mL of Folin–Ciocalteu reagent and 5.0 mL of 20% sodium carbonate Na_2_CO_3_ solution were added, made up to the mark with deionized water and incubated in 21 °C in the absence of light for 60 min., then the absorbance was measured in a spectrophotometer (UV-VIS UV-6100A, Metash Instruments Co., Ltd., Beijing, China) at a wavelength of λ = 750 nm. The determination was performed in 9 independent repetitions, and after accounting for the applied dilutions, the results were calculated on the basis of the calibration curve (y = 2.1297x + 0.1314, R^2^ = 0.9994) for gallic acid as a reference substance and expressed as mg GAE 100g^−1^ D.W. (i.e., mg of gallic acid equivalent per 100 g of dry weight) [27].

### 2.6. Total Flavonoids Measurement

A total of 100 mg of powdered fruit was weighed into a round bottom flask and 10 mL of acetone, 1 mL of hydrochloric acid and 0.5 mL of urotropin (1%) were added. The samples were heated in a hot water bath (30 min.) with a reflux condenser, and the obtained extracts were placed in a 100 mL volumetric flask. The operation was repeated twice, adding 10 mL of acetone each time and heating in the bath for 10 min. The volumetric flask was then charged with acetone (100 mL). Then 20 mL of the extract were transferred to a glass separator, 15 mL of ethyl acetate and 20 mL of deionized water were added and shaken. The separated layer from the bottom was transferred to a new separator, 10 mL of ethyl acetate were added and shaken again. Shaking was repeated 3 times, each time transferring the lower layer to the second separator. At the end of the separation procedure, 50 mL of deionized water was added and shaken one last time. The bottom layer was transferred to a volumetric flask (50 mL) and fill of ethyl acetate was added up to 50 mL. To a small volumetric flask (25 mL) were injected 10 mL of obtained extract, and 2 mL of AlCl_3_ (2%). The sample’s absorbance was measured after 45 min. at λ = 425 nm. The content of total flavonoids (in quercetin equivalents) was calculated from a standard curve prepared for quercetin and expressed as total flavonoids (quercetin equivalents) per 100 g^−1^ D.W. [28].

### 2.7. Total Anthocyanins Measurement

One hundred grams of freeze-dried fruit powder was weighed into a glass beaker and 50 mL of a 1.5 M hydrochloric acid and 95% methanol mixture were added. After carefully mixing, the examined extract was put into a Schott funnel (with a vacuum at 0.1 Bar). Next, another 50 mL of extraction mixture was used for anthocyanin dilution. The filtered extract was moved to a volumetric flask (100 mL) and 10 mL of filtered extract was injected into a smaller volumetric flask (25 mL) and mixed with the extractant. Next, the absorbance was measured at λ = 535 nm. The content of total anthocyanins was calculated with a standard (cyanidin glucoside) and is expressed as mg of CGE 100 g^−1^ F.W. [29]

### 2.8. Identification and Separation of Phenolic Compounds

In order to prepare extracts for assays, 100 mg of freeze-dried blue honeysuckle berries were weighed, 5 mL of 80% methanol was added, vortexed (Micro-Shaker 326 M, Premeo, Poland), incubated in an ultrasonic bath (10 min, 30 °C, 5500 Hz), then centrifuged (10 min, 3780× *g*, 5 °C). The obtained supernatants were centrifuged again (5 min, 31.180× *g*, 0 °C), and then 900 µL of the clear supernatants were transferred to HPLC vials and analyzed.

Identification and separation of phenolic compounds (such as phenolic acids, flavonoids, flavonols) was carried out by the HPLC method, using a Shimadzu kit (USA Manufacturing Inc., Gaithersburg, MD, USA, including two LC-20AD pumps, CBM-20A controller, SIL-20AC column oven, UV/Vis SPD-20 AV spectrometer). The phenolic compounds were separated on a Synergi Fusion-RP 80i Phenomenex column (250 × 4.60 mm), with a flow rate of 1 mL min^−1^. Two gradient phases (acidified with ortho-phosphoric acid, pH 3.0) were used: 10% (*v*/*v*) acetonitrile and ultrapure water (phase A) and 55% (*v*/*v*) acetonitrile and ultrapure water (phase B). The total analysis time was 38 min, with the following phase–time program: 1.00–22.99 min 95% phase A and 5% phase B, 23.00–27.99 min 50% phase A and 50% phase B, 28.00–28.99 min 80% phase A and 20% phase B, and 29.00–38.00 min 95% phase A and 5% phase B. The wavelengths were λ = 250 nm for flavonols and λ = 370 nm for phenolic acids. The phenolic compounds were identified by using 99.9% pure standards (Sigma-Aldrich, Warsaw, Poland) and the specified analysis times for the standards. The standard curves are presented in Appendix A [25].

### 2.9. Identification and Separation of Anthocyanins

The first stage of extraction of samples for anthocyanin analysis was combined with extraction for phenolic acids and flavonols. After extraction with 80% methanol and after the first centrifugation of the extracts (point 2.8.), 2.5 mL of supernatants were taken, 2.5 mL of 10 M hydrochloric acid and 5 mL of 100% methanol were added, shaken gently and then placed in a refrigerator (5 °C, 10 min). Then 1 mL of the clear supernatants were transferred to HPLC vials and analyzed.

The anthocyanins were separated under isocratic conditions at a flow rate of 1.5 mL min^−1^, with one mobile phase consisting of 5% acetic acid, methanol and acetonitrile (70:10:20). The total analysis time was 10 min, at a wavelength of λ = 570 nm. Anthocyanins were identified using 99.9% pure standards (Sigma-Aldrich, Warsaw, Poland) and the specified analysis times for the standards. The standard curves are presented in Appendix A [30].

### 2.10. Allergy Potency Analysis

To determine the allergens (Bet v1 analogues and profilins), an indirect, non-competitive ELISA method was used. Briefly, the Total Protein Extraction Kit for Plant Tissues was used to extract and determine the allergen content. To determine the potential allergen content, the following reagents were used: mouse antibodies against Bet v1 (Dendritics, Brest, France), rabbit antibodies against profilin (Dendritics, Lyon, France), conjugated antibodies against mouse and rabbit immunoglobulins with alkaline phosphatase (Sigma-Aldrich, Warsaw, Poland). The substrate for the alkaline phosphatase was para-nitrophenyl phosphate (Sigma-Aldrich, Warsaw, Poland) and the stop reagent was 3 M NaOH (Sigma-Aldrich, Warsaw, Poland). PBS with 0.1% Tween 20 (Sigma-Aldrich, Warsaw, Poland) was used as a washing agent. The absorbance was measured at λ = 405 nm (Multiscan RC microplate reader, Labsystems, Vantaa, Finland), and the results were calculated using a standard curve prepared with Bet v1 allergen (y = 0.0242x − 0.0697, R^2^ = 0.9923 in the concentration range of 0.5–50 ng mL^−1^) or profilin (y = 0.017x + 0.07415, R^2^ = 0.9942 in the concentration range of 0.5–100 ng mL^−1^). The detection limits for Bet v1 and profiling were 0.88 ng mL^−1^ and 1.2 ng mL^−1^, respectively [31].

### 2.11. Antioxidant Activity Analysis

There are many methods for determining total antioxidant activity [32,33,34]. Therefore, to measure the antioxidant activity, the colorimetric spectrophotometric methods with ABTS^+•^ (2,2′-azino-bis (3-ethylbenzothiazoline-6-sulfonic acid) cation radicals with DPPH (1,1-diphenyl-2-picrylhydrazyl (DPPH) and the ferric reducing/antioxidant power (FRAP) assay were used [35].

For ABTS assay, 0.0384 g of ABTS radical reagent was dissolved in 5.0 mL of deionized water, then 5.0 mL of aqueous potassium persulfate (K_2_S_2_O_8_) (prepared by dissolving 0.026 g in 20 mL of deionized water) was added and the ABTS^+•^ radical solution (10 mL) was incubated at 21 °C in the dark for 12 h and finally diluted with a PBS solution (phosphate-buffered saline), to obtain a blank sample absorbance of 0.700 ± 0.02 at λ = 734 nm. To determine the antioxidant activity, 1.5 mL (diluted with PBS solution) was dispensed into 10 mL glass test tubes, 3.0 mL of ABTS^+•^ cation radical solution (with a predetermined absorbance of 0.700 ± 0.02) was added, incubated at 21 °C for 6 min, and then the absorbance was measured in a spectrophotometer (UV–VIS UV-6100A, Metash Instruments Co., Ltd., Beijing, China) at a wavelength λ = 734 nm.

For DPPH assays, 40 mg L^−1^ of DPPH radical was dissolved in 100% methanol. To determine the antioxidant activity, 20 µL of tested samples extracts were mixed with 3.0 mL of the DPPH solution, incubated at 21 °C for 10 min, and after stabilization, the absorbance was measured in the spectrophotometer (UV–VIS UV-6100A, Metash Instruments Co., Ltd., Beijing, China) at a wavelength λ = 515 nm.

For the FRAP assay, a solution of tripiridyltriazine (TPTZ) (10 mM) and ferric chloride (20 mM) was prepared in 300 mM sodium acetate buffer (pH 3.6) in a ratio of 1:1:10. To determine the antioxidant activity, 20 µL of sample extracts were added to 3.0 mL of TPTZ solution, incubated at 21 °C for 8 min and then the progress of the reduction reaction of ferric tripiridyltriazine (Fe^3+^ TPTZ) to a ferrous form (Fe^2+^) by the antioxidants contained in the extracts was monitored in a spectrophotometer (UV–VIS UV-6100A, Metash Instruments Co., Ltd., Beijing, China) at a wavelength λ = 593 nm. The determination of the antioxidant activity of the tested samples was performed in 9 independent replications for each of the methods used (i.e., ABTS, DPPH and FRAP), and after accounting for the applied dilutions, the results were calculated on the basis of the calibration curves for Trolox as a reference substance and expressed as µM TEAC 1 g^−1^ D.W. Pearson correlation coefficient values among three analytical methods are presented in Appendix A.

### 2.12. Statistical Analysis

The obtained results were analyzed using the statistical program Statgraphics Centurion 15.2.11.0 (StatPoint Technologies, Inc., Warranton, VA, USA). The tables show the average values of nine (n = 9) individual measurements, separately for each year (i.e., 2018 and 2019) for the cultivation system (organic and conventional) and for the three studied cultivars of blue honeysuckle berries (i.e., ‘No 30’, ‘Jolanta’ and ‘Indygo’). Individual blue honeysuckle berry cultivars were represented by six bushes (n = 6) for ‘No 30’, ‘Jolanta’ and ‘Indygo’ cultivar. Two-way analysis of variance with Tukey’s test was performed, and differences between the groups at the level of *p* < 0.05 were considered statistically significant. The standard error (SE) is also given for each mean value presented in the tables. PCA analysis was performed by XLStat (Microsoft Excel trial version, Microsoft^®^ Excel^®^ 2022.4.1, Warsaw, Poland) (accessed on 18 August 2022).

## 3. Results and Discussion

### 3.1. Dry Matter Content

The dry matter contents of the blue honeysuckle fruits from 2018 are shown in Table 1, and those from 2019 are shown in Table 2. During the second year of the experiment organic fruits were characterized by a significantly higher dry matter content (*p* = 0.0059) compared to conventional fruits (13.09 g 100 g^−1^ F.W. vs. 12.33 g 100 g^−1^ F.W.). The higher dry matter content in organic fruit can be explained by the ‘water swelling’ phenomenon. Conventional raspberry tissues collect more water than organic tissues because plants absorb a large amount of water along with mineral fertilizers that are used in conventional farming [36]. The dry matter content in the blue honeysuckle fruit of different cultivars differed significantly during both years of the experiment (*p* = 0.0017 and *p* = 0.0007). In 2018, the highest dry matter content was found in the ‘Indygo’ cv., and in 2019, the highest dry matter content was found in the ‘Jolanta’ cv. The dry matter content was variable in two years of the presented experiment. A similar effect to the obtained was presented by Auzanneau et al. [37]. In 2014, the ‘Indigo Gem’ variety had the highest dry matter content, and in 2015 the ‘Uspiech’ variety, but in 2016 the ‘Indigo Gem’ variety had the highest dry matter content. A similar relationship was observed in other berries—raspberries. In the first year of the experiment, it was found that the ‘Laszka’ variety was characterized by the highest dry matter content, while in the second year of the experiment the highest dry matter content was found in the ‘Glen Fine’ variety [15]. Among the three examined apricot cultivars, in the first year of the experiment it was found that the ‘Orange Red’ cv. was characterized by the highest dry matter content, while in the second experimental year, the ‘Harcot’ cv. was characterized by the highest dry matter content. However, in the second year of the experiment, these differences were not statistically significant [25].

### 3.2. Polyphenols Content

Blue honeysuckle berries are a rich source of polyphenolic bioactive compounds, such as polyphenols, flavonoids and anthocyanins. The total polyphenols (by Folin–Ciocalteu method), flavonoids (by Christ–Müller method), and anthocyanins (by Fulecki and Francis method), measured using the spectrophotometric methods, are presented in Table 1.

As it results from our research (Table 1), blue honeysuckle berries differed in terms of the total polyphenols, total flavonoids and total anthocyanins content (measured using colorimetric, spectrophotometric methods) in the second growing season, while in the first season (2018) higher levels of these bioactive ingredients were found in conventional cultivation than in organic cultivation. In the second growing season (2019), we observed a reverse tendency, i.e., the total polyphenols, flavonoids and anthocyanins content was significantly higher in fruits from organic cultivation (by 13.9%, 24.2% and 15.8%, respectively) than the conventional one for each of the three tested varieties. The observed differences probably resulted from different annual climatic conditions, such as humidity, sunlight and, above all, temperature, determining (for example) the flowering time [3,6]. According to the literature, differences in flowering and then fruiting and ripening may affect the chemical composition of the fruit [3,38]. Moreover, blue honeysuckle is a long-lived plant, bearing fruit for up to 30 years, and the obtained yield increases over time, reaching its maximum values after about 8–15 years [39].

In terms of cultivars, in both growing seasons, the ‘Indygo’ had the significantly highest concentration of the tested bioactive compounds (Table 1) compared to the ‘No 30’ and ‘Jolanta’ cultivar; in the first season (2018) it contained 1.9 and 1.4 times more total polyphenols, 1.7 and 1.5 times more flavonoids and 3.1 and 1.5 times more anthocyanins. In the next season (2019), we observed even greater differences in the content of these components, i.e., the ‘Indygo’ cultivar contained 2.5 and 2.3 times more total polyphenols, 2.3 and 2.4 times more flavonoids and 3.1 and 2.8 times more anthocyanins than the ‘No 30’ and ‘Jolanta’ cultivars, respectively. According to the literature, the variety and genetic features of blue honeysuckle have a significant impact on the chemical composition of the fruit, including the content of polyphenolic bioactive ingredients [1,2,3,7,31,37,40], which was confirmed by the results of our research on these compounds.

The polyphenol contents of the blue honeysuckle fruits, measured by HPLC method, are presented in Table 2 and Table 3. Identification of individual phenolic compounds is also presented in Figure 1 and Figure 2 (chromatograms).

The use of synthetic plant protection products (pesticides) and mineral fertilizers is prohibited in organic farming systems. In such situations, organic plants start to defend themselves by production of polyphenols, which are called “natural pesticides” [39,41]. In contrast, organic fertilizers such as manure and compost are widely used, which supports biodiversity. It also contributes to the increased synthesis of polyphenolic compounds in organic plants. This cultivation approach is largely associated with a lower supply of readily absorbable mineral nitrogen in the soil. In crops for which nitrogen is readily available (conventional), plants primarily produce compounds with high nitrogen contents. However, when the availability of nitrogen is limited (in organic farming), plants produce compounds with high carbon content and secondary metabolites such as polyphenolic compounds [15,42].

Polyphenolic compounds are also produced by plants for defense purposes in states of increased biotic or abiotic stress, and their production is a natural defense mechanism of plants as a result of stress factors related to, e.g., drought, frost and high light radiation. They can also be called ‘natural pesticides’ because they protect plants from biotic factors such as fungi, bacteria, viruses and pests. Therefore, in organic crops that do not use synthetic plant protection products, plants produce more polyphenolic compounds to help with natural protection [15,43,44].

In 2019, organic fruits were also characterized by significantly higher individual phenolic acids, such as chlorogenic, caffeic and *p*-coumaric acid, but conventional fruits contained significantly more gallic acid. ‘Indygo’ cv. had higher caffeic and *p*-coumaric acid contents in 2018. However, in 2019, ‘No 30’ cv. contained more chlorogenic and caffeic acid, and ‘Indygo’ cv. contained significantly more gallic acid.

### 3.3. Flavonoids Content

In 2019, organic blue honeysuckle berries were characterized by a significantly higher content of individual flavonols, namely quercetin-3-*O*-glucoside, kaempferol-3-*O*-glucoside, myricetin and quercetin, as well individual anthocyanins, such as cyanidin-3-*O*-glucoside, than conventional blue honeysuckle berries. This observation confirms the hypothesis that blue honeysuckle has a strong health-promoting effect, due to the high concentration of phenolic bioactive ingredients.

Of all the examined cultivars, ‘Indygo’ was characterized by the highest significant contents of quercetin-3-*O*-rutinoside, kaempferol-3-*O*-glucoside, and cyanidin-3-*O*-glucoside in 2018 and 2019. However, the study performed by Ochmian et al. [37] obtained the following results: 68.13–154.65 mg 100 g^−1^ D.W., total flavonols and 1061.09–1754.43 mg 100 g^−1^ D.W. of total anthocyanins as well as 163.80–222.49 of total phenolic acids mg 100 g^−1^ D.W. Rop et al. [45] compared the fruit from 12 blue honeysuckle cultivars. The total phenolic and flavonoid contents were measured by the Folin–Ciocalteu method. The highest contents of phenolics were recorded in the ‘Zolushka’ cultivar, with values of 6615.38 mg of total polyphenols 100 g^−1^ D.W. and 2937,72 mg 100 g^−1^ D.W. of total flavonoids, respectively. All varieties of haskap berries were cultivated in identical conditions and in the same locations. Therefore, the authors suggest that the variability in the content of polyphenols is the result of variety diversity, which is quite typical for this species.

In a study conducted by Rupasinghe et al. [8], the total phenolic content and total flavonoid content of three blue honeysuckle cultivars, i.e., ‘Borealis’, ‘Indigo Gem’ and ‘Tundra’, were evaluated. The results indicated that the ‘Borealis’ cv. possessed the highest total phenolic contents (4560.44 mg gallic acid 100 g^−1^ D.W.), specifically for total flavonoids (5123.0 mg quercetin 100 g^−1^ D.W.), among the tested cultivars. The total phenolic contents were measured by the Folin–Ciocalteu method as well as total flavonoids by the aluminum chloride spectrophotometric method. Authors described differences in the content of flavonoids, especially anthocyanins, in different haskap berries as related to the differences in the morphological structure and the location of the fruit on the plant. The cultivar ‘Borealis’ was characterized by fruits protected from direct sunlight. The authors suggest that the ‘Indigo Gem’ and ‘Tundra’ cv. had exposed fruit and that most of the anthocyanins were used to reduce UV radiation stress. The contents of phenolic acids and flavonoids in berries from ten different cultivars and clones of blue honeysuckle in a study conducted by Kaczmarska et al. [2] were measure as 142.85 mg of the total phenolic acids, expressed as caffeic acid equivalents per 100 g^−1^ D.W. and 238.82 mg of total flavonoids, expressed as quercetin equivalents per 100 g^−1^ D.W., respectively. The authors concluded that the different bioactive compounds concentration in the examined blue honeysuckle cultivars was strongly related to the genetic diversification of the plants, even though the plants belonged to the same species. Similar results were observed in our experiment. In our experiment, the examined haskap berries differed significantly in terms of the content of bioactive compounds. This was observed especially with the ‘Indigo’ cv. compared to others. This might well be the effect of differentiation between varieties (Table 1).

Databases were used for the study conducted by Oszmiański et al. [5]. Literature sources reported that blue honeysuckle berries are a rich source of phenolic compounds such as phenolic acids as well as anthocyanins, proanthocyanidins and other flavonoids, which display potential health-promoting effects. The potential use of blue honeysuckle berries in preventing chronic diseases such as diabetes mellitus, cardiovascular diseases and cancer seems to be related to their phenolic contents. Celli et al. [46] also reviewed results from recent studies on blue honeysuckle extracts. The total phenolic contents in blue honeysuckle berries ranged from 1025.6 to 8366.3 mg of gallic acid equivalents 100 g^−1^ D.W. as well as 9523.8 mg of cyanidine equivalents per 100 g^−1^ D.W. In our study, the examined blue honeysuckle contained two times higher concentration of total polyphenols, expressed as gallic acid equivalents and ten times less of total anthocyanins, express as cyaniding glucoside equivalents. However, Kucharska et al. [9] reported higher contents of these compounds in different blue honeysuckle cultivars and genotypes as follows: 1101.2–4800.0 mg 100 g^−1^ D.W. for total anthocyanins, 991.9–4082.1 mg 100 g^−1^ D.W. for cyanidin-3-*O*-glucoside, 12.96–171.1 mg 100 g^−1^ D.W. peonidin-3-O-glucoside and 10.76–204.7 mg 100 g^−1^ F.W. for cyanidin-3-*O*-rutinoside. The contents of total anthocyanins, cyanidin-3-*O*-glucoside, peonidin-3-*O*-glucoside and cyanidin-3-*O*-rutinoside in blue honeysuckle berries in the study conducted by Ochmian et al. [37] ranged from 692.1 to 1040.1 mg 100 g^−1^ D.W. for total anthocyanins, from 598.6 to 904.83 mg 100 g^−1^ D.W. for cyanidin-3-*O*-glucoside, from 21.83 to 36.33 mg 100 g^−1^ D.W. for peonidin-3-*O*-glucoside and from 7.32 to 55.38 mg 100 g^−1^ D.W. for cyanidin-3-*O*-rutinoside.

### 3.4. Antioxidant Activity Properties

Antioxidant activity is an excellent example of a functional benefit that plant polyphenols can deliver. Fruits with a high polyphenol concentration are characterized by a higher antioxidant activity as measured in Trolox equivalents [47,48]. Notably, organic farm management can affect the level of polyphenols and their antioxidant power. In experiments on strawberry and raspberry fruits, when organic production methods are used, the fruits always contain more total polyphenols [14,15]. Some results are contrary to those obtained by the authors. Organic apples are characterized by a lower polyphenol level than conventional apples [49].

The higher antioxidant activity was reflected by the higher polyphenol content in blue honeysuckle. We also found a strong correlation between total polyphenol content and antioxidant activity for organic fruits during both experimental years (Table 4). Plants produce polyphenol compounds as an effect of environmental stress reactions. Many experiments confirm this phenomenon [50,51,52]. Some phenolic acids are synthesized by plants as a reaction to insect attack, and some are synthesized as a reaction to abiotic stress conditions such as chemical pesticides. The use of artificial chemical compounds can generate chemical plant responses in stress protein synthesis [53,54]. In our experiment we found this type of plant reaction. In both years, we found a strong correlation among the polyphenol concentration, allergy proteins and Bet v1 concentration (Table 4). According to the literature and our observations, we conclude that artificial plant stressors used in conventional blue honeysuckle cultivation generate a stronger level of plant stress than biotic and abiotic factors characteristic in the organic environment. Of course, basic plant reactions still lead to polyphenol synthesis in plants according to different stress reactions. We observed these plant reactions in 2018 over a broad range but during the next experimental year they covered a narrower range (Table 1 and Table 2). On the one hand, a higher polyphenol concentration in fruits is recommended as a healthy characteristic, but we must remember that it could lead to higher profilins and Bet v1 concentrations in fruits.

### 3.5. Allergenic Potency

Our research showed significant differences in the content of potentially allergenic substances (i.e., Bet v1 and profilins) in the three tested blue honeysuckle berries cultivars (Table 1 and Table 2). The highest concentration of Bet v1 in both growing seasons and profilins was detected in the ‘Indigo’ cultivar, characterized by the highest concentration of polyphenolic compounds. On the other hand ‘No 30’ cultivar contained the lowest concentration of these potentially allergenic proteins, both in 2018 and in 2019, respectively, for Bet v1 and profilins.

The presence of these compounds is bad news for individuals who suffer from intolerances and food allergies. In our first experiment on organic and conventional apricots, we observed similar plant reactions. The fruits with higher polyphenol concentrations showed higher profilins and Bet v1 concentrations [25]. To understand the dependence between bioactive compounds and plant stress reactions, a principal component analysis (PCA) of blue honeysuckle cultivated by organic and conventional methods over two years was performed. PCA analysis is one of the methods of multidimensional dependency analyses used in descriptions to examine the quality and relationship between the features [55,56]. This type of analysis reduces a large amount of data to a small group of linear combinations of related variables. The basis of dependence is correlation. The results showed a high overall variation rate as explained by PC1 and PC2, which were 62.09% in 2018 and 61.63% in 2019, respectively (Figure 3). We found a strong link between polyphenol compounds and allergenic proteins. In 2018, the results for the organic ‘Indigo’ cultivar were close to those of the conventional cultivar, but this latter cultivar correlates with many more phenolics than the organic cultivar. In 2019, another cultivar ‘Indigo’ under conventional production, was positively correlated with phenolics and allergenic compounds. As shown on the graph in 2018, only ‘Jolanta’ fruits were located in different, completely separate areas. This arrangement suggests a complete chemical dissimilarity between the examined fruits. In Figure 3A we can see that the allergenic compound profiles (in the positive direction) were separated from the anthocyanin compounds. However, it was observed that there was a relationship between some phenolic acids (caffeic, *p*-coumaric) and flavonoids (kaempferol-3-*O*-glucoside and quercetin). These dependencies were particularly strong for the ‘Indigo’ cv. for both types of production. Figure 3B showed that there was a similar relationship between allergenic compounds, phenolic acids and selected flavonoids. However, it is worth emphasizing that these factors were located in the PC2 negative direction. A strong correlation was observed for ‘No 30’ cv. for both types of production.

## 4. Conclusions

There are major changes in the chemical composition of blue honeysuckle berries over the course of production in organic systems. Variant environmental stresses and agricultural practices affect the synthesis of polyphenols, profilins and Bet v1 in plants. For consumers who suffer from food intolerance and food allergies, these compounds could generate some problems.

The choice of an appropriate cultivar also has a significant impact on the polyphenol and allergic agents content of blue honeysuckle berries. The presented experiment was conducted using HPLC and Elisa test methods. As our two year research showed, contrary to the best bioactive properties of the ‘Indigo’ cv. (3241.92 mg and 3787.22 mg of total polyphenols per 100 g D.W.), with the highest content of potentially allergenic substances (108.92 ng g^−1^ and 139.27 ng g^−1^ D.W. of Bet v1), the lowest content of Bet v1 and profilins in both 2018 and 2019 seasons was found in the ‘No 30’ cv. (95.41 ng 141.78 ng per g^−1^ D.W. of Bet v1 and 3.51 µg and 5.28 µg per g^−1^ D.W.), with the lowest total polyphenols content (1684.52 mg and 1508.91 mg per 100 g^−1^ D.W.) among the studied blue honeysuckle cultivars.

The obtained results are unique on a global scale because blue honeysuckle is still a relatively new cultivated plant, especially in Poland, and information about its allergenic potency and food safety is insufficient and sometimes even contradictory. This is why further research on potentially allergic and possible anti-allergic properties of blue honeysuckle fruit from organic farming is needed. Further experiments and analyses are necessary to confirm our two year experiment as well as to eliminate randomness connected with climatic data. On the other hand, future experiments should be focused on different cultivars’ properties, comparing organic and conventional production.

## Figures and Tables

**Figure 1 molecules-27-06083-f001:**
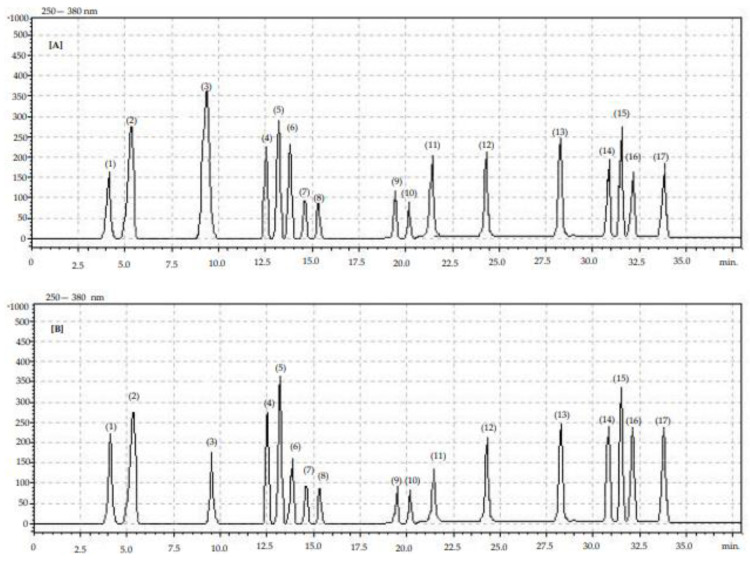
Chromatogram showing retention times for identified phenolic compounds in organic (**A**) and conventional (**B**) blue honeysuckle fruits: (1) gallic acid, (2) phenol unknown, (3) chlorogenic acid, (4) phenol unknown (5) caffeic acid, (6) phenol unknown, (7) phenol unknown, (8) phenol unknown, (9) quercetin-3-*O*-rutinoside, (10) p-coumaric acid, (11) phenol unknown, (12) kaempferol-3-*O*-glucoside, (13) myricetin, (14) luteolin, (16) quercetin, (17) kaempferol.

**Figure 2 molecules-27-06083-f002:**
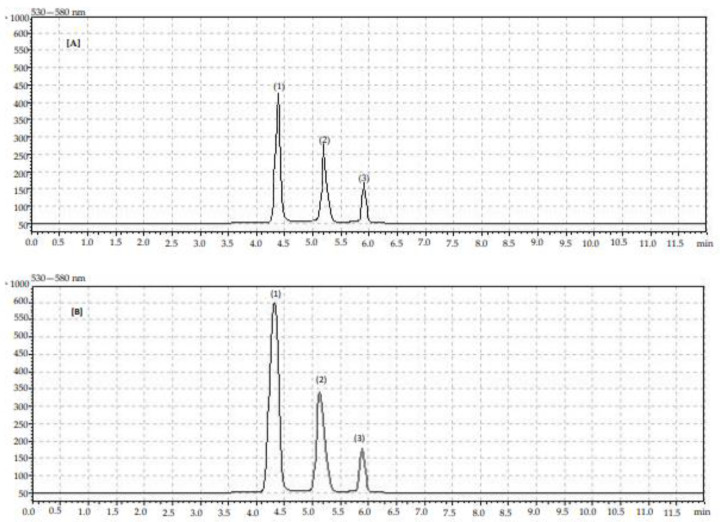
Chromatogram showing retention times for identified anthocyanin compounds in organic (**A**) and conventional (**B**) blue honeysuckle fruits: (1) cyanidin-3-*O*-glucoside, (2) cyanidin-3-*O*-rutinoside, (3) peonidin-3-*O*-glucoside.

**Figure 3 molecules-27-06083-f003:**
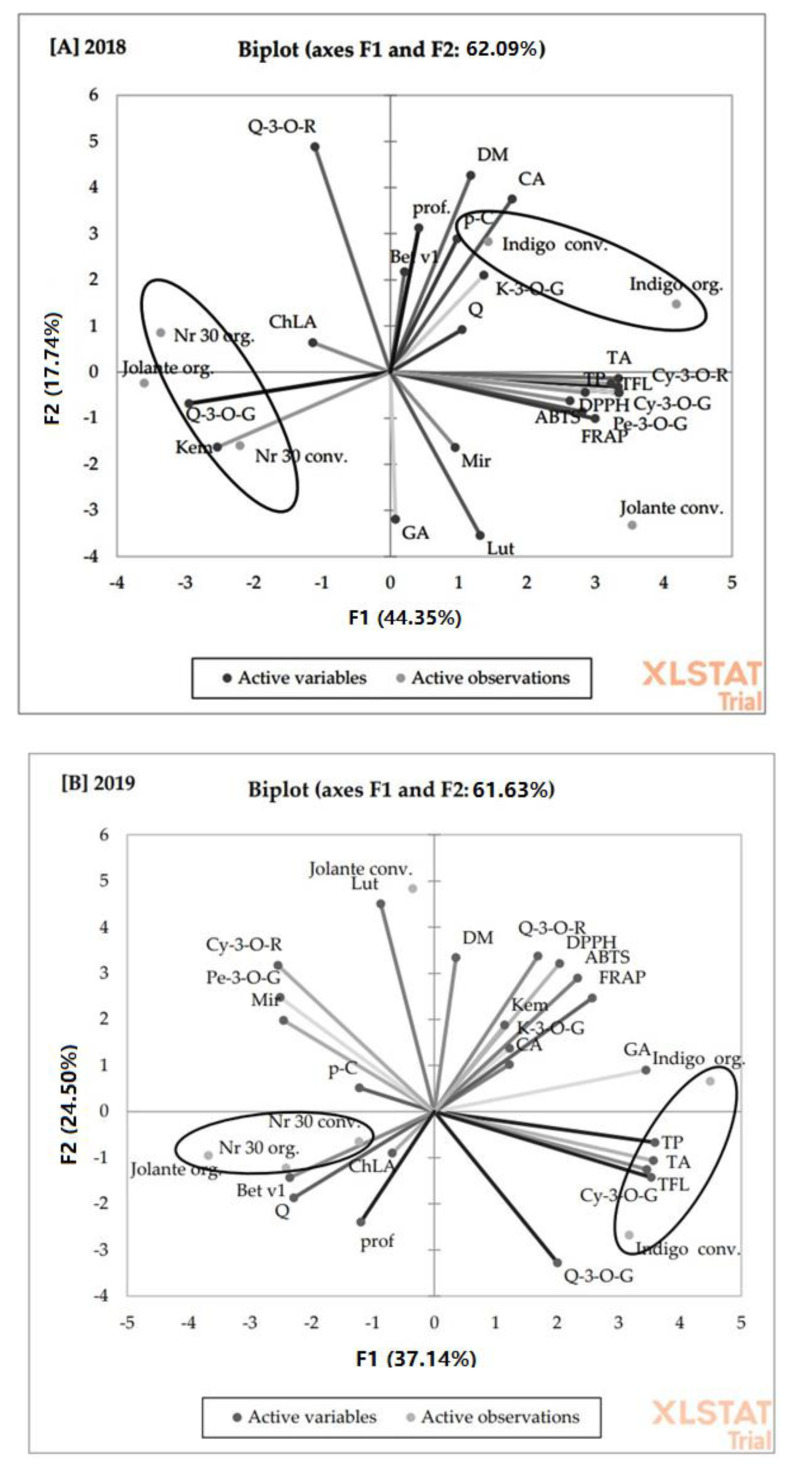
PCA analysis showing the relationship between the chemical composition and allergy potency of organic and conventional blue honeysuckle in both (**A**) 2018 and (**B**) 2019, dry matter (D.M.), total polyphenols (TP), gallic acid (GA), chlorogenic acid (ChLA), caffeic acid (CA), *p*-coumaric acid (p-C), total flavonoids (TFL), quercetin-3-*O*-rutinoside (Q-3-O-R), quercetin-3-*O*-glucoside (Q-3-O-G), kaempferol-3-*O*-glucoside (K-3-O-G), myricetin (Mir), luteolin (Lut), quercetin (Q), kempferol (Kem), total anthocyanins (TA), cyanidin-3-*O*-glucoside (Cy-3-O-G), peonidin-3-*O*-glucoside (Pe-3-O-G), cyanidin-3-*O*-rutinoside (Cy-3-O-R), antioxidant activity (AA), Bet v1 (Bet v1), profilins (prof).

**Table 1 molecules-27-06083-t001:** The content of total polyphenols (by Folin–Ciocalteu method), total flavonoids (by Christ–Müller method), total anthocyanins (by Fulecki and Francis method).

2018	Total Polyphenols mg GAE g^−1^ D.W. *	Total Flavonoids mg QE 100 g^−1^ D.W. **	Total Anthocyanins mg CGE 100 g^−1^ D.W. ***
organic	2118.19 ± 398.71 ^B^	1009.20 ± 199.91 ^B^	815.73 ± 213.26 ^B^
conventional	2726.97 ± 248.95 ^A^	1342.45 ± 83.82 ^A^	1024.96 ± 127.47 ^A^
*p*-value	<0.0001	<0.0001	<0.0001
Nr 30	1684.52 ± 99.14 ^b^	921.05 ± 74.20 ^b^	443.04 ± 50.49 ^b^
Jolanta	2341.30 ± 529.02 ^b^	1040.93 ± 250.90 ^b^	941.53 ± 221.46 ^b^
Indigo	3241.92 ± 223.59 ^a^	1565.50 ± 109.52 ^a^	1376.46 ± 141.60 ^a^
*p*-value	<0.0001	<0.0001	<0.0001
**2019**			
organic	2414.66 ± 395.32 ^A^	1253.76 ± 190.23 ^A^	994.22 ± 185.03 ^A^
conventional	2225.32 ± 304.86 ^B^	1192.57 ± 173.09 ^B^	860.12 ± 161.17 ^B^
*p*-value	0.0005	0.030	<0.0001
Nr 30	1508.91 ± 60.30 ^b^	858.65 ± 46.01 ^b^	538.36 ± 17.01 ^b^
Jolanta	1663.84 ± 71.80 ^b^	827.43 ± 36.48 ^b^	583.28 ± 24.09 ^b^
Indigo	3787.22 ± 127.42 ^a^	1983.41 ± 35.64 ^a^	1659.86 ± 49.78 ^a^
*p*-value	<0.0001	<0.0001	<0.0001

* GAE—gallic acid equivalents, ** QE—quercetin equivalents, *** CGE—cyanidin glucoside equivalents. Results express as mg 100 g^−1^ D.W. (dry weight). Data are presented as the mean ± SE (standard error) with ANOVA *p*-value. Means in column followed by the same letter are not significantly different at the 5% level of probability (*p* < 0.05), A,B values for different cultivation system, a,b values for different cultivars.

**Table 2 molecules-27-06083-t002:** The content of dry matter in (g 100 g^−1^ F.W.), polyphenols (mg 100 g^−1^ D.W.), antioxidant activity (μM TEAC g^−1^ D.W.), Bet v1 (ng g^−1^ D.W.) and profilins (μg g^−1^ D.W.) in examined blue honeysuckle fruits in 2018.

Compounds/Examined Combination	Cultivation Method		Cultivar			*p*-Value	
	Organic	Conventional	Nr 30	Jolanta	Indigo	Cultivation Method	Cultivar
dry matter	14.55 ^1^ ± 0.51 ^A2^	14.66 ± 0.87 ^A^	14.09 ± 0.88 ^a^	12.93 ± 0.30 ^a^	16.80 ± 0.34 ^b^	N.S. ^3^	0.0038
gallic acid	9.70 ± 0.54 ^A^	11.21 ± 0.67 ^A^	10.31 ± 1.00 ^a^	11.14 ± 0.59 ^a^	9.92 ± 0.69 ^a^	N.S.	N.S.
chlorogenic acid	29.64 ± 4.06 ^A^	25.81 ± 1.18	38.34 ± 3.17 ^a^	20.18 ± 1.35 ^b^	24.66 ± 0.58 ^b^	<0.0001	<0.0001
caffeic acid	10.54 ± 1.83 ^B^	15.46 ± 2.86 ^A^	8.95 ± 0.69 ^b^	7.25 ± 0.65 ^b^	22.79 ± 1.97 ^a^	<0.0001	<0.0001
p-coumaric acid	3.02 ± 0.20 ^A^	3.00 ± 0.18 ^A^	3.01 ± 0.29 ^ab^	2.64 ± 0.16 ^b^	3.37 ± 0.12 ^a^	N.S.	<0.0001
quercetin-3-*O*-rutinoside	122.46 ± 1.39 ^A^	98.50 ± 8.60 ^B^	114.02 ± 4.79 ^b^	91.56 ± 10.73 ^b^	125.85 ± 1.78 ^a^	<0.0001	<0.0001
quercetin-3-*O*-glucoside	46.98 ± 8.28 ^B^	50.96 ± 11.21 ^A^	80.71 ± 6.53 ^a^	40.86 ± 9.55 ^b^	25.34 ± 5.76 ^c^	0.0460	<0.0001
kaempferol-3-*O*-glucoside	122.64 ± 26.27 ^A^	30.98 ± 2.45 ^B^	39.99 ± 3.82 ^c^	55.23 ± 13.16 ^b^	135.20 ± 39.45 ^a^	<0.0001	<0.0001
myricetin	13.31 ± 1.09 ^B^	37.12 ± 2.11 ^A^	22.44 ± 2.84 ^b^	28.54 ± 5.27 ^a^	24.66 ± 6.68 ^ab^	<0.0001	0.0080
luteolin	5.82 ± 0.23 ^B^	7.81 ± 1.07 ^A^	5.41 ± 0.19 ^b^	9.42 ± 1.17 ^a^	5.63 ± 0.28 ^b^	<0.0001	<0.0001
quercetin	1.65 ± 0.07 ^B^	3.05 ± 0.33 ^A^	1.77 ± 0.04 ^b^	2.46 ± 0.23 ^a^	2.82 ± 0.60 ^a^	<0.0001	<0.0001
kempferol	10.38 ± 1.34 ^B^	12.27 ± 2.66 ^A^	18.01 ± 2.27 ^a^	10.01 ± 1.56 ^b^	5.94 ± 0.49 ^c^	0.0004	<0.0001
cyanidin-3-*O*-glucoside	499.60 ± 124.01 ^B^	722.12 ± 102.04 ^A^	277.96 ± 22.91 ^c^	668.06 ± 169.88 ^b^	886.58 ± 56.13 ^a^	<0.0001	<0.0001
peonidin-3-*O*-glucoside	48.55 ± 9.34 ^B^	67.72 ± 6.11 ^A^	35.95 ± 3.54 ^c^	59.93 ± 12.01 ^b^	78.53 ± 4.16 ^a^	<0.0001	<0.0001
cyanidin-3-*O*-rutinoside	174.15 ± 17.87 ^A^	172.37 ± 15.40 ^A^	110.37 ± 3.07 ^b^	208.10 ± 10.27 ^a^	201.32 ± 12.03 ^a^	N.S.	<0.0001
ABTS	624.90 ± 35.55 ^B^	676.31 ± 11.64 ^A^	635.20 ± 18.10 ^b^	616.02 ± 41.41 ^b^	700.60 ± 27.95 ^a^	<0.0001	<0.0001
FRAP	585.30 ± 34.66 ^B^	630.12 ± 14.50 ^A^	578.13 ± 15.39 ^b^	587.45 ± 41.33 ^b^	657.54 ± 29.21 ^a^	<0.0001	<0.0001
DPPH	661.18 ± 38.71 ^B^	720.77 ± 9.06 ^A^	673.56 ± 18.41 ^b^	646.40 ± 44.24 ^b^	752.97 ± 26.11 ^a^	<0.0001	<0.0001
Bet v1	106.76 ± 3.37 ^A^	101.51 ± 0.71 ^B^	95.41 ± 4.92 ^b^	108.77 ± 2.32 ^a^	108.92 ± 1.05 ^a^	<0.0001	<0.0001
profilins	3.10 ± 0.18 ^B^	6.07 ± 0.11 ^A^	3.51 ± 0.20 ^b^	3.19 ± 0.19 ^b^	7.05 ± 0.16 ^a^	<0.0001	<0.0001

^1^ Data are presented as the mean ± SE with ANOVA *p*-value; ^2^ Means in rows followed by the same letter are not significantly different (*p* < 0.05), A,B values for different cultivation system, a–c values for different cultivars ^3^ N.S.—not significant statistically.

**Table 3 molecules-27-06083-t003:** The content of dry matter in (g 100 g^−1^ F.W.), polyphenols (mg 100 g^−1^ D.W.), antioxidant activity (μM TEAC g^−1^ D.W.), Bet v1 (ng g^−1^ D.W.) and profilins (μg g^−1^ D.W.) in examined blue honeysuckle fruits in 2019.

Compounds/Examined Combination	Cultivation Method		Cultivar			*p*-Value	
	Organic	Conventional	Nr 30	Jolanta	Indygo	Cultivation Method	Cultivar
dry matter	13.09 ^1^ ± 0.23 ^A2^	12.33 ± 0.35 ^B^	11.97 ± 0.25 ^b^	13.45 ± 0.27 ^a^	12.71 ± 0.40 ^b^	0.0060	0.0007
gallic acid	14.56 ± 1.67 ^B^	16.17 ± 1.27 ^A^	10.47 ± 0.27 ^c^	14.81 ± 1.01 ^b^	20.81 ± 0.34 ^a^	0.0001	<0.0001
chlorogenic acid	35.99 ± 4.94 ^A^	31.33 ± 1.44 ^B^	46.58 ± 3.86 ^a^	24.47 ± 1.64 ^b^	29.92 ± 0.71 ^b^	<0.0001	<0.0001
caffeic acid	7.83 ± 0.56 ^A^	6.89 ± 0.26 ^B^	8.10 ± 0.27 ^a^	6.36 ± 0.36 ^b^	7.62 ± 0.71 ^ab^	0.0014	0.0001
p-coumaric acid	3.14 ± 0.03 ^A^	2.60 ± 0.11 ^B^	3.05 ± 0.06 ^a^	2.85 ± 0.09 ^b^	2.70 ± 0.21 ^b^	<0.0001	0.0002
quercetin-3-*O*-rutinoside	136.60 ± 10.84 ^A^	135.79 ± 19.29 ^A^	100.62 ± 13.71 ^b^	154.40 ± 22.43 ^a^	153.56 ± 9.71 ^a^	N.S. ^3^	<0.0001
quercetin-3-*O*-glucoside	81.16 ± 3.73 ^A^	74.04 ± 12.60 ^B^	61.16 ± 3.21 ^b^	62.80 ± 8.87 ^b^	108.85 ± 7.86 ^a^	<0.0001	<0.0001
kaempferol-3-*O*-glucoside	11.02 ± 0.82 ^A^	8.16 ± 0.45 ^B^	9.62 ± 1.04 ^a^	8.78 ± 0.50 ^b^	10.36 ± 1.20 ^a^	0.0320	<0.0001
myricetin	9.10 ± 0.58 ^A^	8.56 ± 0.46 ^B^	7.98 ± 0.13 ^b^	11.01 ± 0.22 ^a^	7.49 ± 0.10 ^b^	<0.0001	0.0009
luteolin	1.91 ± 0.01 ^B^	1.93 ± 0.10 ^A^	1.88 ± 0.01 ^b^	2.11 ± 0.09 ^a^	1.77 ± 0.07 ^b^	0.0002	<0.0001
quercetin	79.05 ± 6.05 ^A^	69.77 ± 13.31 ^A^	111.58 ± 5.96 ^a^	62.06 ± 9.60 ^b^	49.60 ± 2.37 ^c^	N.S.	<0.0001
kempferol	14.38 ± 0.09 ^B^	16.67 ± 0.68 ^A^	14.30 ± 0.13 ^b^	16.27 ± 0.80 ^a^	16.00 ± 0.81 ^a^	0.0001	0.0022
cyanidin-3-*O*-glucoside	610.08 ± 223.46 ^A^	498.83 ± 17833 ^B^	228.84 ± 5.22 ^b^	35.51 ± 2.72 ^c^	1399.02 ± 63.10 ^a^	<0.0001	<0.0001
peonidin-3-*O*-glucoside	84.69 ± 19.76 ^A^	78.58 ± 19.31 ^A^	56.15 ± 3.43 ^b^	161.96 ± 4.94 ^a^	26.80 ± 1.22 ^c^	N.S.	<0.0001
cyanidin-3-*O*-rutinoside	160.90 ± 21.76 ^A^	161.44 ± 35.70 ^A^	150.21 ± 12.25 ^b^	266.00 ± 21.33 ^a^	67.31 ± 2.80 ^c^	N.S.	<0.0001
ABTS	572.25 ± 23.15 ^B^	652.16 ± 11.17 ^A^	592.22 ± 24.73 ^b^	603.97 ± 36.28 ^b^	640.43 ± 12.11 ^a^	<0.0001	<0.0001
FRAP	554.74 ± 27.57 ^B^	624.29 ± 9.85 ^A^	572.36 ± 27.06 ^b^	568.94 ± 33.47 ^b^	627.26 ± 17.98 ^a^	<0.0001	<0.0001
DPPH	621.82 ± 20.46 ^B^	696.09 ± 13.58 ^A^	649.77 ± 17.02 ^b^	651.11 ± 39.48 ^b^	675.98 ± 11.18 ^a^	<0.0001	<0.0001
Bet v1	142.21 ± 3.37 ^A^	133.41 ± 0.71 ^B^	141.78 ± 4.92 ^a^	138.39 ± 2.32 ^ab^	139.27 ± 1.05 ^b^	<0.0001	<0.0001
profilins	5.37 ± 0.18 ^A^	5.27 ± 0.11 ^A^	5.28 ± 0.20 ^a^	5.45 ± 0.19 ^a^	5.24 ± 0.16 ^a^	N.S.	N.S.

^1^ Data are presented as the mean ± SE with ANOVA *p*-value; ^2^ Means in rows followed by the same letter are not significantly different (*p* < 0.05), A,B values for different cultivation system, a–c values for different cultivars ^3^ N.S.—not significant statistically.

**Table 4 molecules-27-06083-t004:** Pearson’s coefficient (R^2^) between anti-allergenic potency, antioxidant activity and total polyphenols in organic and conventional blue honeysuckle fruits in 2018 and 2019.

2018	Organic Berries	Conventional Berries
	polyphenols	
Bet v1	0.472	0.987
*p*-value	0.0041	<0.0001
**2019**	organic berries	conventional berries
	polyphenols	
Bet v1	0.581	0.872
*p*-value	0.017	0.0001
**2018**	organic berries	conventional berries
	polyphenols	
profilins	0.128	0.921
*p*-value	N.S.	0.0041
**2019**	organic berries	conventional berries
	polyphenols	
profilins	0.505	0.911
*p*-value	0.0032	0.0001
**2018**	organic berries	conventional berries
	polyphenols	
antioxidant activity (ABTS)	0.886	0.857
*p*-value	<0.0001	0.0005
**2019**	organic berries	conventional berries
	polyphenols	
antioxidant activity (ABTS)	0.925	0.834
*p*-value	<0.0001	0.0002

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
