# Peer review of "Polyphenols Content, Antioxidant Properties and Allergenic Potency of Organic and Conventional Blue Honeysuckle Berries"

_molecules, 2022, doi:10.3390/molecules27186083_

Round 1

Reviewer 1 Report

This manuscript reports the two-year analysis and comparison of the polyphenol content, antioxidant, and alergogenic potentials of three organically and conventionally grown blue honeysuckle berry cultivars. 

An abstract is too long. Based on the journal instruction, the Abstract should be 200 words maximum. Therefore, the Abstract should be more concise.

 Throughout the manuscript (in the Abstract (line 40), in the Introduction (line 106), and the Conclusions (line 536), the authors stated that blue honeysuckle is a "new plant." The authors probably meant "new cultivated plant." Please correct it. What is the status of this fruit under EU regulations?

Please mention previous data on the polyphenol content and bioactivities of studied cultivars. In this context, does this study provide any novelty on summarized literature data by Golba et al. (2020)?

Gołba M, Sokół-Łętowska A, Kucharska AZ. Health properties and composition of honeysuckle berry Lonicera caerulea L. An update on recent studies. Molecules. 2020 Feb 9;25(3):749.

The presentation of the Materials and Methods is adequate and clear to understand. Correct "antioxidant analysis" to "antioxidant activity analysis." Why was only one assay (ABTS) used to evaluate antioxidant activities?

The Tables are understandable, but the Figures' quality needs improvement.

Consider dividing the Results and Discussion section into subsections. Also, there is no need to repeat results from the tables in the text (lines 313 to 315, and lines 487-494). The text should highlight key findings or trends, not repeat previously presented data. Please correct that.

In general, the Discussion of the results is successfully interpreted, and the conclusions are justified by the data.

Author Response

Thank you very much for the review and for your recommendation and suggestions on how to improve our manuscript and increase its quality to the requirements of the “Molecules” journal

Below you can see our replies for all your comments and suggestion. All suggested changes and corrections have been done as well in manuscript text in system track changes.

Comment 1: “An abstract is too long. Based on the journal instruction, the Abstract should be 200 words maximum. Therefore, the Abstract should be more concise.”

Authors’ response: Authors want to apologise. According to Reviewer suggestion Abstract was make shorter. Now is not more than 200 words (199 words).

Comment 2: “Throughout the manuscript (in the Abstract (line 40), in the Introduction (line 106), and the Conclusions (line 536), the authors stated that blue honeysuckle is a "new plant." The authors probably meant "new cultivated plant." Please correct it. What is the status of this fruit under EU regulations?”

Authors’ response: Authors agree with Reviewer suggestion. “New plant” was supposed to suggest that it is a new cultivated plant, especially in Poland on the mass scale. Appropriate correction have been made to the text of the manuscript in all the places indicated. ”new plant” for “new cultivated plant in Poland”

According to information published by the Chief Sanitary Inspectorate, the fruits of the blue honeysuckle, also known as haskap berries, have been officially registered in Poland as edible fruit since 2018. Since this year, they have also been admitted to mass circulation and sale. Similar regulation refers to European Union countries. Blue honeysuckle fruits were officially restarted as “novel food” in 2018. 

Article 3, § 2, lit. a) Regulation of the European Parliament and of the EU Council No. 2015/2283 of 25 November 2015 on “novel foods”, amending Regulation (EU) No. 1169/2011 of the European Parliament and of the Council and repealing EC Regulation No. 258/97 of the European Parliament and of the Council, and Commission Regulation (EC) No. 1852/2001 (OJ EU L 327 of 11 December 2015, p. 1) referred to as Regulation No. 215/2283 “novel food” means food that has not been used for human consumption to a significant extent in the Union before 15 May 1997, irrespective of the dates of accession of the Member States to the Union (…). It may be newly developed or innovative food, or food produced with the use of new technologies and production processes. In addition, the information provided to the Chief Sanitary Inspectorate by the Department of Agricultural Markets (case reference RR.Po.071.4.2017 of March 31, 2017) clearly shows that in Poland cultivars of haskap berries from the turn of the 1980s and 1990s they were found only in amateur cultivation.

Blue honeysuckle fruits were not widely known in Poland and were not sold on the market before May 15, 1997. Taking into account the above, it informs that the blue honeysuckle have been notified to the European Commission as traditional food from third countries, pursuant to the above-mentioned Regulation No. 215/22 83. As evidenced by the relevant information on the website of the European Commission.

Comment 3: “Please mention previous data on the polyphenol content and bioactivities of studied cultivars. In this context, does this study provide any novelty on summarized literature data by Golba et al. (2020)?

Gołba M, Sokół-Łętowska A, Kucharska AZ. Health properties and composition of honeysuckle berry Lonicera caerulea L. An update on recent studies. Molecules. 2020 Feb 9;25(3):749.”

Authors’ response: Authors want to underlined, that in the cited review article by Golba et al. 2018 studies of allergenic compounds and their effects on health are not included. This article takes into account the impact of polyphenolic compounds from haskap berries on the problem with cholesterol and liver function, anti-inflammatory properties and insulin metabolism regulating properties, as well as on antimicrobial properties. The research presented in our manuscript is innovative and unique. They indicate a higher safety of haskap berries when grown organically, especially in terms of problems with food allergies and food intolerance. At the same time, the cultivars examined by the Authors are not included in the article by Golba et al. 2018. This is important information both for consumers and producers of haskap berries.

Comment 4: “The presentation of the Materials and Methods is adequate and clear to understand. Correct "antioxidant analysis" to "antioxidant activity analysis." Why was only one assay (ABTS) used to evaluate antioxidant activities?”

Authors’ response: According to Reviewer suggestion “antioxidant analysis” is corrected into “antioxidant activity analysis”. The authors added the results of analyses of antioxidant activity done by two other methods (FRAP and DPPH). These results were not shown in an previous version of the manuscript. The Authors concluded, that the most popular method of measuring for antioxidant activity is the ABTS method. Now tables present the results of measuring the antioxidant activity done by the use of three methods.

Comment 5: “The Tables are understandable, but the Figures' quality needs improvement.”

Authors’ response: According to Reviewer suggestion resolution of all Figures were corrected in manuscript text.

Comment 6: “Consider dividing the Results and Discussion section into subsections. Also, there is no need to repeat results from the tables in the text (lines 313 to 315, and lines 487-494). The text should highlight key findings or trends, not repeat previously presented data. Please correct that.”

Authors’ response: Authors want to apologise, according to Reviewer suggestion all data values were removed from pointed manuscript places, as well section Results and Discussion was dividing for thematic sub-sections.

Reviewer 2 Report

Title: Polyphenols content, antioxidant properties and allergy potency of organic and conventional blue honeysuckle berries

Comments to authors:

This study focused on comparing the phenolic content, antioxidant properties and allergic characteristics of three blue honeysuckle berries between 2018 and 2019.The work is interesting and valuable to the journal readers, but the experimental content and manuscript expression need to be improved. The followings are some comments and suggestions for authors to consider and improve the manuscript.  

1) The abstract is too general and too long, it is suggested to add specific values of some indicators, such as the content of representative phenolic fractions.

2) It is not appropriate to use only the ABTS clearance rate to represent the antioxidant capacity, and it is suggested to add several more antioxidant indexes to the test.

3) I recommend the results of bioactive compounds, allergy potency and antioxidant activity expressed as dry matter instead of fresh weight.

4) The conclusion section suggests adding some presentation of experimental methods and data.

5) The introduction to polyphenols in Lines 373-388 is suggested to be placed in the introduction.

6) The authors cite numerous references in lines 422-437, but only list the results and do not discuss them.

7) The format of cited reference in the text is incorrect. For instance, Line 215, 2018 should deleted. Line 242, 2020 should deleted.

Author Response

Thank you very much for the review and for your recommendation and suggestions on how to improve our manuscript and increase its quality to the requirements of the “Molecules” journal

Below you can see our replies for all your comments and suggestion. All suggested changes and corrections have been done as well in manuscript text in system track changes.

Comment 1: “The abstract is too general and too long, it is suggested to add specific values of some indicators, such as the content of representative phenolic fractions.”

Authors’ response: Authors want to apologise. According to Reviewer suggestion Abstract was make shorter. Now is not more than 200 words (199 words), as well some representative results for polyphenols and allergens were added to that part of manuscript.

Comment 2: “It is not appropriate to use only the ABTS clearance rate to represent the antioxidant capacity, and it is suggested to add several more antioxidant indexes to the test.”

Authors’ response: Authors agree with Reviewer suggestion, that is why results of analyses of antioxidant activity by two other methods (FRAP and DPPH)  were added into tables in manuscript text. These results were not shown in an previous version of the manuscript. The Authors concluded, that the most popular method of measuring for antioxidant activity is the ABTS method. Now tables present the results of measuring the antioxidant activity done by the use of three methods (ABTS, FRAP and DPPH).

Comment 3: “I recommend the results of bioactive compounds, allergy potency and antioxidant activity expressed as dry matter instead of fresh weight.”

Authors’ response: According to Reviewer suggestion all data about polyphenols, allergy potency and antioxidant activity were expressed in dry matter. All tables and some Figures were corrected in manuscript text.

Comment 4: “The conclusion section suggests adding some presentation of experimental methods and data.”

Authors’ response: According to Reviewer suggestion information about analytical methods and some representative results values were added into Conclusion section.

Comment 5: “The introduction to polyphenols in Lines 373-388 is suggested to be placed in the introduction.”

Authors’ response: According to Reviewer suggestion information about polyphenols function in organic plants was added into manuscript text.

Comment 6: “The authors cite numerous references in lines 422-437, but only list the results and do not discuss them.”

Authors’ response: According to Reviewer suggestion detailed discussion of previous listed result values obtained by other Authors and own result were added to manuscript text (lines 443-466).

Comment 7: “The format of cited reference in the text is incorrect. For instance, Line 215, 2018 should deleted. Line 242, 2020 should deleted.”

Authors’ response: Authors want to apologise. All cited references were corrected according to Molecules Authors Guideline

Reviewer 3 Report

Comments.

This work is fascinating. However, I have made some comments that need to be made to improve the manuscript before it is published. There are a lot of grammatical errors. I have corrected all the grammatical errors in the article and attached them to these comments.

Comment 1

2.3. Preparation of plant material

The freeze dried samples were ground to powder in a laboratory grinder (A-11), sealed, and stored          at -80°C for further testing.

My question is, Was it not sieved for size homogeneity? If yes, what is the mesh size? Please state it.

Comment 2

2.12 Statistical analysis.

Also, write the software for the principal component analysis.

Comment 3

3. Results and discussion

In 2018, 294 the highest dry matter content was found in ‘Indygo’ cv., and in 2019, the highest dry 295 matter content was found in ‘Jolanta’ cv.

My question is. Can you also add literature from other works whose authors also reported similar results?. This will improve the manuscript.

Comment 4

Figure 3. Principal component analysis (PCA).

In the PCA, I suggest you explain more because a lot of information can be explained to support your findings. I can see that are a lot of excellent relationships.

You can look at these articles as a guide.

 https://doi.org/10.1016/j.indcrop.2021.113421

 https://doi.org/10.1016/j.fbio.2021.101044

 https://doi.org/10.1111/1750-3841.15999.

Author Response

Reviewer no. 3

Thank you very much for the review and for your recommendation and suggestions on how to improve our manuscript and increase its quality to the requirements of the “Molecules” journal

Below you can see our replies for all your comments and suggestion. All suggested changes and corrections have been done as well in manuscript text in system track changes.

Comment 1: “2.3. Preparation of plant material. The freeze dried samples were ground to powder in a laboratory grinder (A-11), sealed, and stored at -80°C for further testing. My question is, Was it not sieved for size homogeneity? If yes, what is the mesh size? Please state it.”

Authors’ response:  According to information from Manual the Mill IKA® A-11 basic is equipped with a funnel and sieve with a 0.063 mesh. The freeze-dried material is grounded into a fine powder and sieved automatically. A relevant description has been added to the subsection.

Comment 2: “2.12 Statistical analysis. Also, write the software for the principal component analysis.”

Authors’ response:  Authors want to apologise. In subsection “2.21 Statistical analysis” A relevant description of PCA software has been added

Comment 3: “3. Results and discussion. In 2018, 294 the highest dry matter content was found in ‘Indygo’ cv., and in 2019, the highest dry 295 matter content was found in ‘Jolanta’ cv. My question is. Can you also add literature from other works whose authors also reported similar results?. This will improve the manuscript.”

Authors’ response:  According to Reviewer suggestion relevant references were added to manuscript text as well, the discussion on the variability of the dry matter content in fruit of various cultivars and species in subsequent years of the experiment was complied.

Comment 4: “Figure 3. Principal component analysis (PCA). In the PCA, I suggest you explain more because a lot of information can be explained to support your findings. I can see that are a lot of excellent relationships. You can look at these articles as a guide.

 https://doi.org/10.1016/j.indcrop.2021.113421

 https://doi.org/10.1016/j.fbio.2021.101044

 https://doi.org/10.1111/1750-3841.15999.”

Authors’ response: According to Reviewer suggestion more detailed description and discussion of PCA results has been added to part of the section: “Results and discussion, page 16, lines: 608-631.

Round 2

Reviewer 2 Report

The manuscript was well revised and it could be accepted in its current form.

Author Response

Thank you very much for the review and for your recommendation and suggestions on how to improve our manuscript and increase its quality to the requirements of the “Molecules” journal